# Improved Positive Predictive Performance of *Listeria* Indicator Broth: A Sensitive Environmental Screening Test to Identify Presumptively Positive Swab Samples

**DOI:** 10.3390/microorganisms7050151

**Published:** 2019-05-27

**Authors:** Alan D. Olstein, Joellen M. Feirtag

**Affiliations:** 1Paradigm Diagnostics, Inc., 800 Transfer Road Suite 12, St. Paul, MN 55114, USA; 2Department of Food Science and Nutrition, University of Minnesota, St. Paul, MN 55108, USA; jfeirtag@umn.edu

**Keywords:** food safety, environmental *Listeria*, *Listeria* detection

## Abstract

PDX-LIB, *Listeria* Indicator Broth, was developed as a proprietary sensitive screening test to identify presumptively positive environmental swab samples for *Listeria* sp. The original formulation, while sensitive, initially proved to exhibit acceptable levels of false positive test results. Paradigm Diagnostics has been undertaken to modify the medium formulation to render it more selective while not sacrificing its sensitivity. After identification of a candidate formulation through laboratory studies, a field trial was conducted to validate the test performance parameters, including the true positive frequency and false positive frequency in several different food-processing facilities. Identical swab samples were enriched in both the original medium formulation and the new formulation. Presumptive positive samples were confirmed by plating on selective differential agar and qPCR analysis. The field trial data demonstrate that the new formulation significantly reduces the frequency of false positive samples compared to the original *Listeria* Indicator Broth formulation, without compromising the sensitivity of the original formulation. The new medium formulation resulted in no false positive samples compared to the 54% increased presumptive positive samples obtained with the original medium formulation.

## 1. Introduction

In a risk assessment study, the U.S. Department of Agriculture Food Safety Inspection Service provided the rationale for mandating a national surveillance program for *Listeria* occurrence in USDA-regulated facilities [1]. These new regulations mandated environmental surveillance for the presence of *Listeria* sp. in food processing facilities to minimize the risk of foodborne illness associated with contaminated food. This development impelled many firms, including Paradigm Diagnostics, to develop simple *Listeria* screening tests to enable the growing demand for this test volume [2].

A comprehensive study by the Center for Disease Control in 2012 provided evidence that the implementation of environmental controls in food processing facilities coupled with robust public health monitoring (Pulse Net) helped to reduce the burden of foodborne Listeriosis [3]. Despite these encouraging results, foodborne illnesses due to pathogens, including *Salmonella*, STEC, and *Listeria*, continue to be a challenge in the national food production system [4,5,6]. Figure 1 demonstrates that the frequency of Listeriosis outbreaks in the US has experienced a marked increase in the past few years. Consequently, accurate simple screening methods for foodborne illness pathogens must be available to address the on-going need for facility environmental surveillance.

In this study, we intend to demonstrate that an improved *Listeria* enrichment formulation can help to eliminate uncertainty when screening environmental samples for the presumptive presence of *Listeria* sp. Field trial data collected from eight different food-processing facilities supports the laboratory data, showing that the new formulation, LIB v.2.0, is more accurate than the antecedent test, LIB. Specifically, the false positives observed using LIB were completely eliminated using LIB v.2.0 without a loss of sensitivity for the detection of true *Listeria* positive samples. Appendix A was included to provide detailed location information of where the samples were obtained.

## 2. Materials and Methods

PDX-LIB and *Listeria* Indicator Broth v.2.0 and Securswabs were supplied by Paradigm Diagnostics, Inc. St. Paul, MN. Swabs were collected as duplicates from the same locations in food processing facilities and enriched in 20 mL of either LIB, the original formulation, or LIB v.2.0, the new medium formulation, for 48 h at 37 °C. Blackened samples were streaked onto modified MOX (modified Oxford) medium and incubated for an additional 18 h at 37 °C. The modified MOX medium was prepared by substituting the esculin in the standard MOX formulation with 5 g/L D-arabitol and 0.02 g/L bromcresol purple as the indicator system for *Listeria* sp. [7].

MOX-positive plates were confirmed as *Listeria* sp. by qPCR using primers and probes as detailed in the Food and Drug Administration Bacteriological Assay Manual [8]. Statistical analysis was conducted and pairwise comparisons between pathogen isolation rates using LIB v2.0 and LIB (original formulation) were made using the Mantel-Haenszel chi-square formula for unmatched test portions [9]. A Chi-Square value of less than 3.84 was considered to indicate no significant numerical difference between the two methods being compared. The formula for χ^2^ is
χ^2^= (|a-b|-1)^2^/(a+b)
a = The number of presumptively positive samples using LIB v.2.0.b = The number of presumptively positive samples using LIB.

## 3. Results

A total of 161 samples were obtained from eight different food-processing facilities. Presumptive positive samples were identified and confirmed. Table 1 summarizes the results of field trial samples. Of the 161 environmental samples, LIB v-2.0 yielded 35 presumptive positives, while the original formulation resulted in 55 blackened samples. The 35 LIB v-2.0 samples were confirmed as true positives by plating and PCR analysis.

The LIB (original formulation) results yielded 54 presumptive positives, of which 35 were confirmed. Twenty of the presumptive positive LIB samples were deemed false positives. One hundred and seven of the LIB samples were negative, of which 106 were true negatives. One of the negative LIB samples was deemed a false negative since the duplicate LIB v-2.0 sample yielded a true positive result. Chi square analysis (Χ^2^ = 30.06) of the positives and false positives generated by both sample populations indicated a significant difference at the 95% confidence level.

## 4. Discussion

Listeria environmental screening continues to represent a significant proportion of global *Listeria* testing carried out in the food microbiology laboratory [10]. Accordingly, facile methods to identify presumptively positive environmental samples reduce the cost and time required. Paradigm Diagnostics developed an environmental screening test to identify presumptive positive *Listeria* samples. The method has been shown to be more sensitive than the USDA method [11] and potentially avoids the risk of false negative samples due to the presence of acriflavin in the enrichment medium used by most commercial enrichment media [12].

The data set in Table 1 represent environmental samples from diverse sources of food-processing facilities, Appendix A. The data translate to a sensitivity and specificity for LIB (original formulation) of 97.2% and 86.2%, respectively. In contrast, the sensitivity and specificity data for LIB v-2.0 are 100% and 100%, respectively. The positive predictive values of the respective media are 63% for LIB and 100% for LIB v-2.0.

The field data underscore the substantially better diagnostic performance characteristics of LIB v-2.0 compared with the original LIB formulation. Furthermore, the sensitivity of the new medium appears to be comparable to or better than the original formulation. We had anticipated that the new formulation would exhibit more false negatives since LIB v-2.0 contains higher levels of lithium chloride than LIB. However, we found that the LIB v-2.0 medium exhibited a greater sensitivity, with a value of 100% versus 97.2% for LIB.

This may make sense when one considers that the growth of competitive microflora, particularly *Enterococcus* sp., may inhibit the growth of *Listeria* sp. in the sample. In a recent publication, Hanachi et al. detail the potential to use *Enterococcus* sp., especially *E. faecalis* and *E. faecium*, to control the growth of *Listeria monocytogenes* in food products [13]. In addition to *Enterococcus* sp., many species within the lactic acid bacteria family are capable of producing anti-listerial compounds. The ability of these organisms to compete with *Listeria* sp. resides in their capability to both grow more robustly and produce anti-listerial bacteriocins [14].

Appendix A provides detailed site information from which the samples were obtained at their respective facilities.

In conclusion, we have demonstrated that the new formulation of the environmental *Listeria* screening test, LIB v-2.0, exceeds the performance characteristics of the original formulation, LIB, in comparison field trials. LIB v-2.0 provides a greater accuracy and a higher positive predictive value without sacrificing the test sensitivity.

## Figures and Tables

**Figure 1 microorganisms-07-00151-f001:**
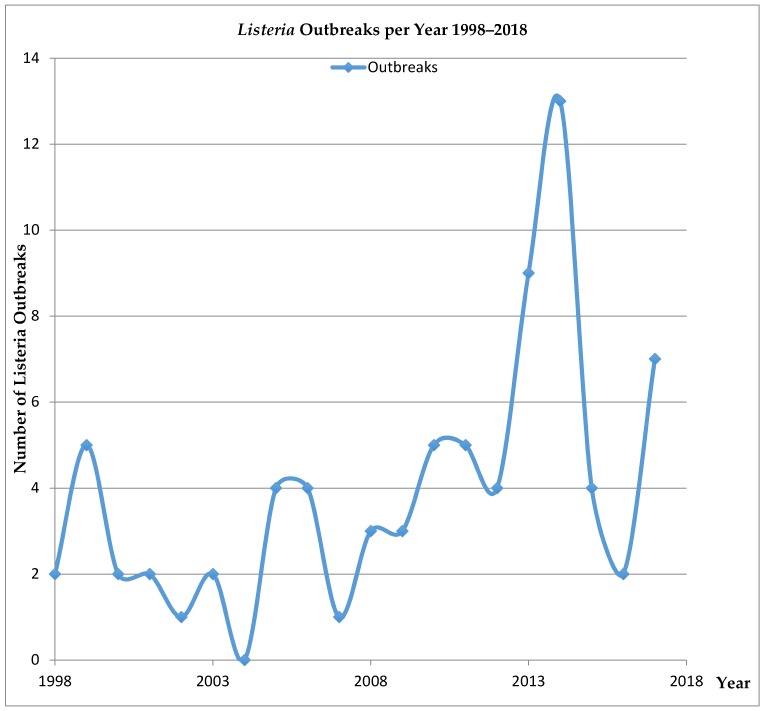
Listeria Outbreaks in the U.S. 1998–2018*. * From the NORS Dashboard Available at https://wwwn.cdc.gov/norsdashboard/. (Accessed on 9 May 2019).

**Table 1 microorganisms-07-00151-t001:** Field Trial Summary.

Medium	Total Samples	Presumptive Positives	Negatives	TP*	TN	FP	FN	Χ^2^
LIB	161	54	106	34	106	20	1	
LIBv-2.0	161	35	126	35	126	0	0	30.06

TP = true positive, TN = true negative, FP = false positive, FN = false negative. *Confirmed using MOX plating and qPCR as described in the US Food and Drug Administration Bacteriological Assay Manual [8].

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
