# Peer review of "Improved Positive Predictive Performance of Listeria Indicator Broth: A Sensitive Environmental Screening Test to Identify Presumptively Positive Swab Samples"

_microorganisms, 2019, doi:10.3390/microorganisms7050151_

Round 1

Reviewer 1 Report

The manuscript presents some interesting results. A few issues, however, need to be addressed:

-     In the abstract section:

The objective is too broad and should be narrowed down to the primary objective. The abstract should include (purpose, methods, scope) and includes part of the results and conclusions of the research and the recommendations of the author. The abstract must be detailed as possible within the word counts limits.

 -  In the introduction section:

I think that the introduction could/should be stronger. The introduction is very broad, I suggest starting with the larger context to engage the reader, followed by a statement of the problem, ending with how you are addressing this problem and what your objective (s).

I would like to see some statistical analysis about food outbreak for Listeria included. A sentence that clearly states the objective (s) of this study at the end of the introduction.

-       In the materials section:

In line 42; which the samples were “enriched in 20 mL of the respective medium”, the authors should be more specific, what is exactly their enriched medium.  

In line 43; I think the authors should express that the meaning of MOX medium (abbreviation) is the Modified Oxford Medium since it was the first usage in the main text.

In line 47; the “positive MOX plates were confirmed as Listeria sp. by qPCR using primers and probes”. I would highly recommend include some figures to address these results.

In line 49; It is unclear to me what is the difference between PDX-LIB and Listeria Indicator Broth version 2.0, and LIB 2.0. Where is the innovation in this study, if they are the same?  

Also, it is unclear to me what is the difference between LIB 2.0 and LIB (original formulation).

-   Later in the discussion section, in line 81, and 84, the author(s) refer to “the new medium”, where's that coming from?

In line 41; the authors mentioned that they were collected samples from “the same locations in food processing facilities”, so how many samples you have been collected? and I recommend that the authors should briefly state the locations that the samples were been collected.

-   Then later at the results section (in line 54, and 55), they have mentioned that 161 samples were obtained from eight different food-processing facilities. It is really confusing, which one is the samples were collected from the same locations in food processing facilities or from eight different food-processing facilities.

The authors mentioned LIB (original formulation) multiple times in the manuscript content. However, it is unclear to the readers what is the component of this Medium

- Appendix A section is too long, I would strongly recommend rewriting or re-organizing this section in a different way. Additionally, there are different highlighted rows with different colors, is these colors has a specific meaning.  

I would recommend that the author(s) may use boldface type, italic type, or underlining, to highlight individual values in tables (this is totally up to the author(s) and to the journal policy). However, any highlighted section must have a supplemental note of explanation and attach the note symbol to the highlighted first value.

Author Response

Dear Reviewer One:

       Thank you for your thoughtful suggestions to improve the manuscript. I have uploaded a revised copy of the manuscript which addresses many of the points you raised.

I have a question regarding your point concerning the statement in L 47 about preparing a chart or table. Are you suggesting a table providing detailed probe/primer sequence? or a flow chart providing the details of the analysis?

Re: L 41 I have enclosed a statement in the introduction bringing the reader reader's attention to the appendix to have detailed information about where the samples were obtained.

Reviewer 2 Report

Authors in the present study present an interesting improvement concerning the sensitivity of identification of Listeria sp. However, the the manuscript does not meet the requirements prior to publication in many ways.

-          Too small introduction, it is not referring even what Listeria sp. is as a foodborne pathogen. Νot all species of Listeria cause listeriosis. Maybe attention should be given to L. monocytogenes. What other diagnostic tools today are used, what does todays' literature have revealed about Listeria sp.pathogenicity or occurance in food products or EU alerts. Enrichment is required in order to cover this study topic.

-          Results is not appropriate for any publication to be presented by one table giving one single information since all other results be presented as Appendix. What about statistical analysis, were they significant? The external validity of these results is not being discussed.

-          Regarding to results and appendix, they need to proceed to implementations in order to well-describe the study and make the paper in a more readable version.

-          Findings and discussion should be endorsed with more references and describe better all the contributions and implications as to the paper results. i.e. limitations regarding possible selection biases with the study sample could be discussed.

Minor comments:

PRX-LIB is not presented in abbreviations. Generally, abbreviations should be avoided in the abstract.

Listeria sp. should be corrected in italics form.

A ‘field trial’ is usually mentioned in agricultural studied.

It is recommended to change ‘foodborne illness’ to foodborne diseases.

In line 81 the reference provided do not fit with the growing demand for new screening test.

Ref 5 needs a date provided.

Ref 12 is not presented in the text.

The manuscript in the present form is only suggested as a short communication letter or a scientific technical report.

Author Response

Dear reviewer two:

        Thank you for your thoughtful critiques. My responses below:

You are correct that not all Listeria sp. are pathogenic to humans. However, environmental screening for the presence of Listeria sp. as an indicator organism of facility hygiene is a widespread practice. The use of Listeria sp. is essentially utilized as an early warning system to indicate a loss of environmental process control in the food industry.

Not  sure the point you're making here, although the data presented in Table 1 summarize the entirety of the results of 161 samples collected from eight food processing facilities. Statistical analysis of the results are incorporated within the summary table. The comparison of presumptive positive samples enriched in LIB and LIBv.2.0 clearly show  statistically significant difference, i.e. that there are high levels of false positive results determined with LIB enrichment vs. LLIB v.2.0.

The appendix was included to provide the reader with detailed location information about where in these facilities the samples were obtained.

Regarding information biases, the study design selected food processing across a broad range of product types to minimize differences with respect to background micro-flora common to different foods. Agreed we did not explicitly state this in the introduction nor discussion. We can certainly put this statement into the manuscript?

Agreed that this manuscript should be submitted as a scientific technical report.

PDX-LIB is defined in abbreviations. It is the products' commercial name.

Listeria sp. corrected.

Field trials can refer to broad range of products not exclusively intended for agricultural use. Lot of new products such electronics, manufactured products are "field tested" before commercial release.

Foodborne illness is common parlance in the United States as well as elsewhere.

Changed reference #10.

Round 2

Reviewer 1 Report

I think the method in this manuscript is significant enough, and properly more appropriate as a note. However, there are a few suggestions for the author (s) to consider:

In line 46 the study objective needs to be more precise; I recommend that to rewrite the objective.

The material and methods need more work, still is not clear what is the significant difference between “LIB or LIB v.2.0 medium”

In line 60 the author mention using qPCR. However, there are no results to support that.

The statistical analysis is not clear I recommend that to rewrite this section again.

In line 97 “medium LIB v-2.0 contains higher levels of selective agents than the original formulation LIB” what type of selective agents.

Author Response

Dear Reviewer One:

Please see revision in Objectives, L. 51-52.

Please see revision in M&M section,L 57-58.

Please see footnote in Table 1.

Please see M&M L. 68-71.

Please see L 107.

Reviewer 2 Report

I would like to thank the authors for taking my previous comments into consideration. I still have some doubts that the specific manuscript in the present form is suggested as a short communication letter or a scientific technical report. However this manuscript is indeed a revised form of its previous submission since the authors proceed to implementations to improve the overall quality.

Author Response

Dear Reviewer Two:

     We agree with your assessment that the submission should qualify as scientific technical note or short paper.
